# CCL2 Inhibition of Pro-Resolving Mediators Potentiates Neuroinflammation in Astrocytes

**DOI:** 10.3390/ijms23063307

**Published:** 2022-03-18

**Authors:** Irene L. Gutiérrez, Fabiana Novellino, Javier R. Caso, Borja García-Bueno, Juan C. Leza, José L. M. Madrigal

**Affiliations:** 1Department of Pharmacology and Toxicology, School of Medicine, Universidad Complutense de Madrid (UCM), Instituto de Investigación Sanitaria Hospital 12 de Octubre (Imas12), Instituto de Investigación Neuroquímica (IUINQ-UCM), Centro de Investigación Biomédica en Red de Salud Mental (CIBERSAM), Avda. Complutense s/n, 28040 Madrid, Spain; irenlo02@ucm.es (I.L.G.); fabnovel@ucm.es (F.N.); jrcaso@med.ucm.es (J.R.C.); bgbueno@med.ucm.es (B.G.-B.); jcleza@med.ucm.es (J.C.L.); 2Institute of Molecular Bioimaging and Physiology (IBFM), National Research Council, Viale Europa, 88100 Catanzaro, Italy

**Keywords:** CCL2, MCP-1, resolvin, LPS, astrocytes, inflammation

## Abstract

The chemokine CCL2 participates in multiple neuroinflammatory processes, mainly through the recruitment of glial cells. However, CCL2 has also been proven to exert different types of actions on these cells, including the modification of their response to inflammatory stimuli. In the present study we analyzed the effect of CCL2 on the resolution of inflammation in astrocytes. We observed that genetic removal of CCL2 increases the expression of the enzymes responsible for the synthesis of specialized pro-resolving mediators arachidonate 15-lipoxygenase and arachidonate 5-lipoxygenase in the brain cortex of 5xFAD mice. The expression of FPR2 receptor, known to mediate the activity of pro-resolving mediators was also increased in mice lacking CCL2.The downregulation of these proteins by CCL2 was also observed in cultured astrocytes. This suggests that CCL2 inhibition of the resolution of inflammation could facilitate the progression of neuroinflammatory processes. The production of the pro-inflammatory cytokine IL-1beta by astrocytes was analyzed, and allowed us to confirm that CCL2 potentiates the activation of astrocytes trough the inhibition of pro-resolving pathways mediated by Resolvin D1. In addition, the analysis of the expression of TNFalpha, MIP1alpha and NOS2 further confirmed CCL2 inhibition of inflammation resolution in astrocytes.

## 1. Introduction

The lack of adequate control of inflammatory responses within the central nervous system (CNS) is known to be the main cause of the damage characteristic of many neurodegenerative diseases. Among the different mediators responsible for the propagation of inflammatory signals, chemokines constitute one of the most relevant groups. Chemokines are a kind of cytokine that can affect the behavior of some cells by various means, but their main way of interacting is the activation of specific G-protein-coupled receptors located on target cells. The activation of these chemokine receptors results in the movement of cells towards the areas where chemokines accumulate, acting in this way as beckoning chemical signals. Due to this effect, chemokines are key agents in many processes in which cell migration is involved, with the inflammatory response being among the most studied ones.

Multiple studies indicate that the chemokine monocyte chemoattractant protein 1 (MCP-1 or CCL2) contributes to the progression of neuroinflammatory processes and the neurodegeneration associated with them [1]. In agreement with this, our previous studies allowed us to observe that the genetic suppression of CCL2 in the 5xFAD mouse model of Alzheimer’s disease reduces the accumulation of amyloid β plaques, the production of pro-inflammatory mediators and the neuronal damage [2].

Most CCL2 actions are mediated by the activation of its specific receptor, known as CCR2. CCR2 is a chemokine receptor consisting of seven transmembrane domains. Its extracellular N-terminal domain specifically binds different chemokines, with CCL2 being the most potent activator of CCR2. The activation of CCR2 initiates a series of intracellular signaling pathways leading to a chemotactic response in the cells expressing this receptor. In the CNS, CCR2 is present in astrocytes, endothelial cells, microglia, and neurons [3]. The accumulation of CCL2 is associated with the progression of neuroinflammatory processes [4]. This could be the result of different effects mediated by CCR2 activation including the attraction of glial cells to inflammation sites which may contribute to the potentiation of this response. In this way, indirect neurotoxic effects of CCL2 have been attributed to the activity of infiltrated monocytes and microglia [5] stimulated by this chemokine. However, CCL2 does not seem to activate these cells directly, and genetic CCR2 deletion has been proven to facilitate the progression of neurodegeneration in different mouse models of Alzheimer’s disease [6,7]. Therefore, CCL2 regulation of neuroinflammation in the CNS seems to be a complex process in which CCR2-independent effects could play a relevant role.

Based on this, we decided to analyze whether the accumulation of CCL2 modifies the processes involved in the resolution of inflammation in glial cells, which seem to play a key role in the progression of Alzheimer’s disease [8]. For this purpose, we analyze here the regulation by CCL2 of the Resolvin D1 (RvD1) pathway which is known to play a key role in the resolution of inflammation within the CNS and in the progression of neurodegenerative disorders [9].

## 2. Results

### 2.1. CCL2 Deletion Increases FPR2, 15-LOX and 5-LOX mRNA Expression in 5xFAD Mice

RT-PCR studies allowed us to quantify the expression of N-formyl peptide receptor 2 (FPR2), key RvD1 receptor [10], in the brain cortices obtained from WT and 5xFAD mice. This comparison did not allow us to detect significant differences between the two types of mice. However, FPR2 mRNA concentrations were increased in samples obtained from CCL2-KO and 5xFAD/CCL2-KO mice (Figure 1).

In addition to FPR2, we also analyzed the expression of arachidonate 15-lipoxygenase (15-LOX) and arachidonate 5-lipoxygenase (5-LOX), enzymes responsible for the production of lipid metabolites including RvD1 [11]. In this way, we observed that the absence of CCL2 in 5xFAD mice increases the expression of 15-LOX and 5-LOX (Figure 1). Nevertheless, the expression of these enzymes was not increased in CCL2-KO mice. Therefore, the potential regulation of 15-LOX and 5-LOX expression by CCL2 may only take place in the presence of inflammatory stimuli like the ones present in the brains of 5xFAD mice.

### 2.2. CCL2 Prevents the Expression of RvD1 Synthesizing Enzymes and Receptors

The increased expression of inflammation-resolution mediators observed in the brain cortex from CCL2-KO mice indicates that the activity of this chemokine may inhibit their production. To confirm the regulatory effect of CCL2 on the expression of FPR2, 5-LOX and 15-LOX, we analyzed the expression of these proteins in vitro. For these studies, we decided to use astrocytes because, although different cell types take part in the inflammatory response within the brain cortex, astrocytes are the main source of CCL2 in the brain [12]. This allowed us to analyze how both resting and activated astrocytes, isolated from other cells, respond to a transient increase of CCL2 concentration in their environment. 

In addition, the use of rat cells allowed us to quantify the regulation of G protein-coupled receptor 32 (GPR32), which, along with FPR2, is one of the two receptors known for RvD1. While present in humans, GPR32 was detected as a pseudogene in rodents [13] but is RNA sequence was predicted for rat (NCBI reference sequence XM_039097471.1) and its production was later detected in rat conjunctival goblet cells [14].

The expression of 15-LOX, 5-LOX, GPR32 and FPR2 was increased when astrocytes were stimulated with LPS. However, a pre-treatment of astrocytes with CCL2 prevented the effect of LPS on the expression of all the lipoxygenases and receptors analyzed (Figure 2), confirming this way the inhibitory action of CCL2 on the expression of these mediators.

### 2.3. CCL2 Potentiates the Production of Inflammatory Markers Induced by LPS in Astrocytes

According to our results, CCL2 inhibition of the expression of the lipoxygenases and receptors analyzed could prolong the resolution of the inflammatory response and sustain the production of inflammatory and neurotoxic agents. To confirm this, we first analyzed the production of the cytokine interleukin (IL)-1β due to its well characterized role as a mediator responsible for the propagation of pro-inflammatory signals and neuronal damage. This way, we observed that 48 h after the addition of LPS, IL-1β expression was still greatly activated, and this effect was increased in those cells treated with CCL2 (Figure 3A). 

The effect of CCL2 on IL-1β could be validated by the quantification of the concentration of IL-1β released to the culture medium by ELISA (Figure 3B). In addition, Western blot analyses allowed us to further confirm the effect of CCL2 by measuring the accumulation of IL-1β inside the astrocytes (Figure 3C,D).

To determine if CCL2′s elevation of IL-1β levels was due to its inhibition of FPR2 and GPR32 receptors, we analyzed the effect of RvD1, the specific ligand for FPR2 and GPR32. In this way, we observed that CCL2 completely blocked the inhibition of IL-1β expression exerted by RvD1 (Figure 3E).

In addition to IL-1β, we analyzed the expression of other markers of astrocytic activation and neuroinflammation such as tumor necrosis factor-α (TNFα), macrophage inflammatory protein-1 α (MIP1α) and nitric oxide synthase 2 (NOS2). Similarly to what we found in the case of IL-1β, the expression of these markers was induced by LPS and CCL2 also potentiated this effect (Figure 3F).

## 3. Discussion

Our previous studies indicate that CCL2 plays a relevant role in the progression of Alzheimer’s disease [2]. This is supported by studies performed on human samples in which high CCL2 plasma levels are associated with increased disease severity [15]. Based on this observation, and on the impaired resolution of inflammation known to occur in Alzheimer’s disease, we decided to analyze whether CCL2 modulates the resolution of inflammation in the brain. For this purpose, we focused on the RvD1 pathway, as it is one of the most relevant ones in the resolution of inflammation, although the contribution of other specialized pro-resolving mediators cannot be discarded. This way, our in vivo results demonstrate that, in the absence of other stimuli, neither CCL2 suppression nor the genetic modifications present in 5xFAD mice, alter the expression rate of the RvD1 synthesizing enzymes 15-LOX and 5-LOX in the brain cortex. However, when CCL2 is removed from 5xFAD mice, a large induction of both lipoxygenases takes place. This suggests that, in inflammatory conditions such as the ones that partially resemble Alzheimer’s disease in 5xFAD mice, CCL2 may exert a repressive effect on the expression of these enzymes.

Therefore, according to our in vivo data, the restriction of 15-LOX and 5-LOX expression by CCL2, and the resulting lower levels of one of their main metabolites such as RvD1, would facilitate the extension of inflammatory processes due to the pro-resolving actions of this mediator which largely depend on the activation of FPR2 receptor [16]. In support of this hypothesis, we found that CCL2-KO mice have higher cortical concentrations of FPR2 mRNA which could compensate for lower levels of its ligand.

The following in vitro analyses performed confirm our hypothesis. The astrocyte cultures data demonstrate that the induction of resolution mediators, namely the lipoxygenases 15-LOX and 5-LOX as well as the RvD1 receptors FPR2 and GPR32, which takes place in astrocytes after their activation [17], is inhibited by CCL2. This could extend the inflammatory process and, therefore, facilitate the accumulation of pro-inflammatory cytokines and other neurotoxic substances. In accordance with this, we found that CCL2 prevents RvD1 inhibitory effect and potentiates LPS activation of astrocytes.

It should be remarked that the GPR32 mRNA analyzed here relates to the predicted rat pseudogene sequence; therefore, additional analyses are still necessary to confirm that the changes detected correspond with the GPR32 gene.

Astrocytes [18] and microglia [19] can react to many different stimuli, generating several diverse functional responses depending on the stimulus perceived by these cells. Interestingly, for this study, LPS treatment of cultured microglia has been demonstrated to upregulate different mediators of the resolution pathway analyzed here, such as 5-LOX and FPR2, while the exposure to Aβ1-42 peptide did not have these effects [20]. In addition, Aβ has been shown to directly interact with FPR2 receptor [21]. For these reasons, while the analysis of Aβ effects constitutes and interesting study, we believe that the use of LPS, and therefore, the generation of an inflammatory response through the activation of toll-like receptors [22] is the most useful model to analyze CCL2 regulation of the inflammation-resolution process.

The activation of GPR32 and FPR2 receptors restricts the recruitment of immune cells and the subsequent amplification of their response [23], limiting in this way the extension of the inflammation and the cellular damage it may cause. In the CNS, the activation of FPR2 has been demonstrated to reduce the neuronal damage in Alzheimer’s disease, ischemia or sepsis [24]. In astrocytes, the suppression of FPR2 reduces their resistance to bacterial injuries [25].

In the conditions used here, CCL2 causes a large inhibition of the expression of the specialized pro-resolution markers analyzed. This is accompanied by an increase in the production of IL-1β and in the expression of the pro-inflammatory mediators TNFα, MIP1α and NOS2. Therefore, we conclude that CCL2 potentiation of neuroinflammation in astrocytes is due, at least in part, to its inhibition of resolution mediators.

## 4. Materials and Methods

### 4.1. Mouse Models and Brain Samples Collection

All experimental protocols adhered to the guidelines of the Animal Welfare Committee of the Universidad Complutense of Madrid, Spain (PROEX 174/18), and according to European Union laws (2010/63/EU).

Wild-type (WT) C57BL/6, 5xFAD (strain B6.Cg- Tg (APPSwFlLon, PSEN1* M146L*L286V) 6799Vas/Mmjax), and CCL2KO (B6.129S4-Ccl2tm1Rol/J) mice were obtained from The Jackson Laboratory. These mice were maintained for over 10 generations on a C57BL6 background. 5xFAD mice express 3 mutant forms of human APP under the control of the murine Thy-1 promoter, and the murine presenilin-1 (PSEN1) with the M233T and L235P familial AD (FAD)-linked mutations expressed under the control of the mouse PSEN1 promoter. Hemizygous 5xFAD mice were crossed with homozygous CCL2KO mice. The resultant 5xFAD+/−/CCL2+/− mice were backcrossed with CCL2KO mice to generate 5xFAD+/CCL2KO mice. For these studies, 5-month-old male WT, heterozygous 5xFAD, homozygous CCL2KO, and 5xFAD/CCL2KO (heterozygous 5xFAD and homozygous CCL2KO) were used. Six or more mice were included in each experimental group. Mice were housed up to 4 per cage in a controlled 12:12 h light/dark cycle, with food provided ad libitum. All efforts were made to minimize animal suffering and to reduce the number of animals used.

The animals were killed by terminal anesthesia using sodium pentobarbital (250 mg/kg ip Vetoquinol^®^, Madrid, Spain) and subjected to transcardial perfusion with saline. The brains were removed, cortical areas excised from the brain, snap-frozen and kept at −80 °C.

### 4.2. Cell Cultures

Rat cortical astrocytes were obtained as described previously [26]. Briefly, 1-day-old Wistar rats (Harlan) were used to prepare primary mixed glial cultures; microglia were detached by gentle shaking after 11–13 days in culture, astrocytes were prepared by mild trypsinization of the remaining cells and consisted of 95% astrocytes as determined by staining for GFAP, and <5% microglial as determined by staining with the specific marker OX-42. All cultures were free of mycoplasma contamination, which was tested using mycoplasma detection kits (Biotools). CCL2 and LPS were dissolved in water and RvD1 was dissolved in EtOH 0.037%. Controls received the equivalent amount of vehicle.

### 4.3. mRNA Analysis

Total cytoplasmic RNA was prepared using TRIzol reagent (Thermo Fisher Scientific, Carlsbad, CS, USA), aliquots converted to cDNA using random hexamer primers and SuperScript^®^ Reverse transcriptase (Thermo Fisher Scientific), and mRNA levels estimated by quantitative real-time PCR (QPCR). PCR conditions were 35 cycles at 95 °C for 10 s, annealing at 60 °C for 15 s, and extension at 72 °C for 30 s followed by 5 min at 72 °C in a Corbett Rotorgene. Reactions were carried out in the presence of Sybr Green (Biotools). Relative mRNA levels were calculated by comparison of take-off cycles and normalized to values for GAPDH measured in the same sample. The following primers were used (forward/reverse, 5′ to 3′):
Mouse FPR2:CTGGGCTCAAACTGATGAAGA/CGTAAAGGACGGCTGGAATTAMouse 15-LOX:GGGACAATGGACACCGTTATTA/CCAGGTACTGCTGACTACAAAG Mouse 5-LOX:TCAAGCAGCACAGACGTAAA/GCCATCCAGTAGCTCGTAATCRat 15-LOX:CTCCACTACAAGACCGACAAAG/GTGCATTAGGAACCCAGTAGAARat 5-LOX:CTGGTAGCCCATGTGAGGTT/GCACAGGGAGGAATAGGTCARat GPR32:CCTTTCTGGTTCTCACCTTCTT/GTGATGGCCTGTCTCTCTTTCRat FPR2:CGCTGTCAAGATCCACAGAA/CTCCAAACTGGAAGGCAGAGRat IL-1β:ACCTGCTAGTGTGTGATGTTCCCA/AGGTGGAGAGCTTTCAGCTCACATRatMIP-1α:CAGAACATTCCTGCCACCTGCAAA/AGGAATGTGCCCTGAGGTCTT TCARat TNFα:CTGGCCAATGGCATGGATCTCAAA/AGCCTTGTCCCTTGAAGAGAA CCTRat NOS2:AGCACATTTGGCAATGGAGACTGC/AGCAAAGGCACAGAACTGAGG GTARat/mouse GAPDH:TGCACCACCAACTGCTTAGA/GGCATGGACTGTGGTCATGAG

### 4.4. Western Blot

Samples containing equal amounts of protein were subjected to sodium dodecyl sulfate-polyacrylamide gel electrophoresis on 10–20% SDS-polyacrylamide gels. The proteins were then transferred to poly- vinyl difluoride membranes, which were blocked with 5% bovine serum albumin in Tris-buffered saline containing 0.1% Tween-20 for 1 h and incubated overnight at 4 °C with primary antibodies against IL-1β (ab254369, Abcam, Cambridge, UK) or β-actin (A5441, Sigma, Madrid, Spain). This was followed by incubation with anti-IgG-horseradish peroxidase-labeled secondary antibodies for 1 h at room temperature and subsequent detection with an enhanced chemiluminescence detection kit (ECL, Sigma, Madrid, Spain). Blots were imaged using an Odyssey^®^ Fc System (Li-COR Biosciences). Several exposition times were analyzed to ensure the linearity of the band intensities. The resulting bands were quantified by densitometry (Image J^©^). All densitometries are expressed in arbitrary units (AU).

### 4.5. IL-1β Measurement

Protein levels in the culture medium were detected using specific enzyme-linked immunosorbent assays (ELISA), carried out according to the manufacturer’s instructions (Thermo Fisher Scientific).

### 4.6. Statistical Analysis

All experiments were performed at least in triplicate. Data were analyzed by one-way analysis of variance (ANOVA), followed by Tukey’s multiple comparison tests and *p* values < 0.05 were considered significant. The numbers of samples (*n*) indicated in the figure legends correspond to biological replicates and animals for in vitro and in vivo experiments, respectively.

## Figures and Tables

**Figure 1 ijms-23-03307-f001:**
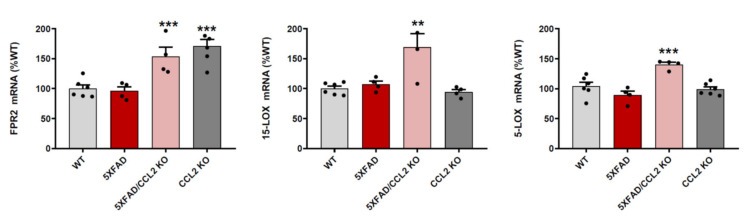
CCL2 deficiency increases the expression of pro-resolution mediators. FPR2, 15-LOX and 5-LOX mRNA concentrations were analyzed in brain cortex samples from WT, 5xFAD, CCL2-KO and 5xFAD/CCL2-KO mice. Data are means ± SE of *n* = 6 replicates per group. *** *p* < 0.001, ** *p* < 0.01 vs. WT.

**Figure 2 ijms-23-03307-f002:**
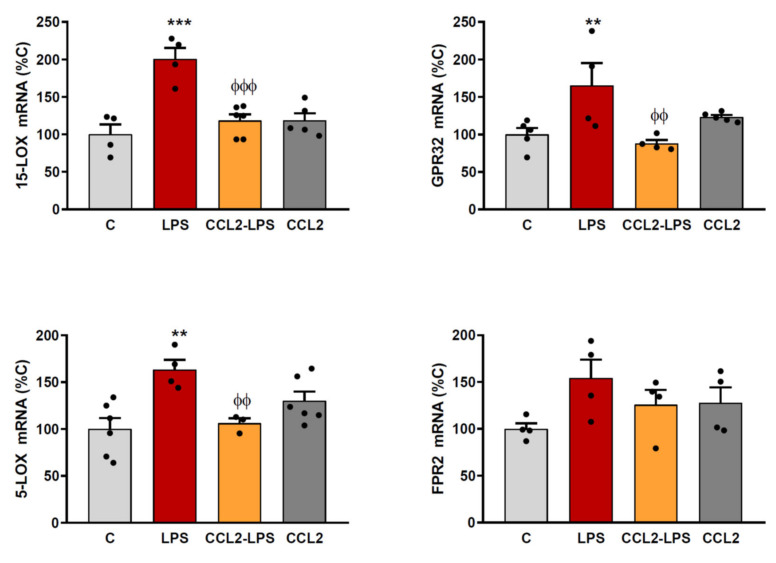
CCL2 prevents the expression of pro-resolution markers. Astrocyte cultures were incubated with or without CCL2 (100 ng/mL). Twenty-four hours later, the whole medium was replaced in all wells, maintaining CCL2 where it was used for pre-treatment and adding LPS (0.1 g/mL) to indicated groups. Six hours later, RNA was isolated and 15-LOX, 5-LOX, FPR2 and GPR32 mRNA levels determined by RT-PCR. Data are means ± SEM of n = 6 replicates per group. *** *p* < 0.001, ** *p* < 0.01 vs. C. ^ϕϕϕ^
*p* < 0.001, ^ϕϕ^
*p* < 0.01 vs. LPS. ϕ.

**Figure 3 ijms-23-03307-f003:**
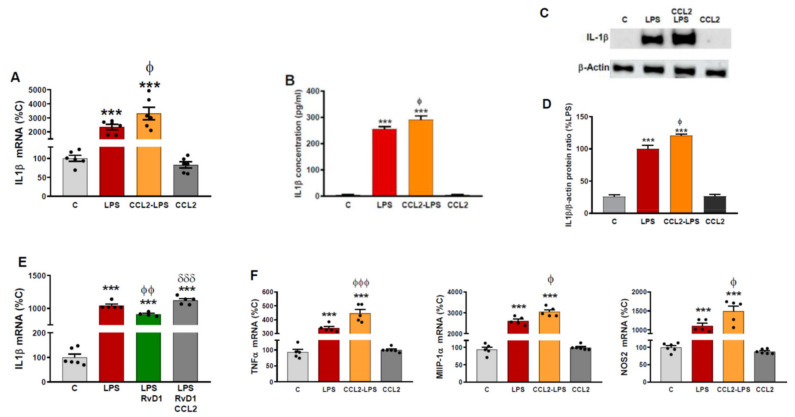
CCL2 induces production of astrocytic inflammatory markers. (**A**) Astrocyte cultures were incubated with or without CCL2 (100 ng/mL). Twenty-four hours later, the whole medium was replaced in all wells, maintaining CCL2 where it was used for pre-treatment and adding LPS (0.1 g/mL) to indicated groups. Forty-eight hours later, RNA was isolated and IL-1β mRNA levels determined by RT-PCR. Data are means ± SEM of n = 6 replicates per group. *** *p* < 0.001 vs. C-C. ^ϕ^
*p* < 0.05 vs. LPS. (**B**) IL-1β concentration in the media was measured by ELISA. Data are means ± SEM of n = 6 replicates per group. *** *p* < 0.001 vs. C. ^ϕ^
*p* < 0.05 vs. LPS. (**C**) Cytosolic lysates were examined for the presence of IL-1β and β-actin protein by Western blot analysis. The gels shown are representative of experiments done on three separate astrocyte preparations. (**D**) Densitometric analysis of the bands. AU: arbitrary units relative to C. *** *p* < 0.001 vs. C. ^ϕ^
*p* < 0.05 vs. LPS. (**E**) Astrocyte cultures were incubated with vehicle (EtOH 0.037%), RvD1 100 nM (dissolved in EtOH 0.037%) or RvD1+CCL2 (100 ng/mL). Twenty-four hours later, the whole medium was replaced in all wells, maintaining RvD1 and CCL2 where it was used for pre-treatment and adding LPS (0.1 g/mL) to all groups except the control one. Forty-eight hours later, RNA was isolated and IL-1β mRNA levels determined by RT-PCR. Data are means ± SEM of n = 6 replicates per group. *** *p* < 0.001 vs. C. ^ϕϕ^
*p* < 0.01 vs. LPS. ^δδδ^
*p* < 0.001 vs. LPS+RvD1. (**F**) Astrocyte cultures were incubated with or without CCL2 (100 ng/mL). Twenty-four hours later, the whole medium was replaced in all wells, maintaining CCL2 where it was used for pre-treatment and adding LPS (0.1 g/mL) to indicated groups. Forty-eight hours later, RNA was isolated and TNFα, MIP1α and NOS2 mRNA levels determined by RT-PCR. Data are means ± SEM of n = 6 replicates per group. *** *p* < 0.001 vs. C-C. ^ϕϕϕ^
*p* < 0.001, ^ϕ^
*p* < 0.05 vs. LPS.

## Data Availability

Not applicable.

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
