# Peer review of "CCL2 Inhibition of Pro-Resolving Mediators Potentiates Neuroinflammation in Astrocytes"

_ijms, 2022, doi:10.3390/ijms23063307_

Round 1

Reviewer 1 Report

The paper attempts to elucidate the pro-inflammatory role of CCL2 under specific stimuli conditions. The current study is a follow up on the mechanisms involving the reduced neuroinflammation in CCL2KO-5X FAD mice as previously observed by the authors. My comments are as follows:

  1. Proof reading for grammatical integrity of the sentences.
  2. Represent the qRT PCR data as an individual sample graph to give an idea of each sample value.
  3. Please comment on the cells primarily expressing CCL2 receptors and their role in neuroinflammation.
  4. When astrocytes are subjected to another inflammatory stimuli such as IL-1beta or oxidative stress, does CCL2 pretreatment cause a similar trend in the expression of these genes? In other words, is the observed trend specific to LPS. Neuroinflammation, oxidative stress, microglial dysfunction all contribute to the pathology observed in 5X FAD mice. Any comment on the role of CCL2 on these intrinsic contributing factors.
  5. In the in vitro astrocyte treatment, it is not clear whether the media was replaced in all wells at 24hr irrespective of any treatment.
  6. What are the RvD1 expression levels in brain lystes of WT, 5X FAD, CCL2 KO, and CCL2 KO-5XFAD?

Reviewer 2 Report

In their paper, Gutiérrez et al. provide results suggesting that CCL2 conterbalance the inflammation-resolving effects of resolvin D1, a lipid mediator. Although results are of potential interest, several major drawbacks preclude publication of this paper in IJMS.

Indeed results are way too preliminary for a research paper ; only 3 figures and a couple of molecules examined by ELISA, WB and/or RT-PCR  are not enough to draw any robust conclusion. This all the more that 1 figure deals with investigations performed on whole brain tissue derived from mice while the 2 others presenty results obtained on cultured rat astrocytes. As stated by the author, GPR32, one of the Resolvin D1 receptor, is a pseudogene in rodents. So what is the point of assessing its level of expression in rat astrocytes? One would have expected that authors take advantage of CCL2 KO mice to perform analyses on murine astrocytes from WT vs CCL2 KO mice. In addition the use of 5XFAD mice in Fig 1 is questionable  since it is not a model of acute neuroinflammation. What is the link between assessing inflammation in the brain of CCL2KO x 5xFAD mice and measuring the effects of LPS + CCL2  on the synthesis of IL-1 beta in rat astrocytes? Why not using amyloid beta instead of LPS as a pro-inflammatory trigger (as for e.g. in PMID: 28266714 or PMID: 15755672)?

Ideally, the authors should extend the results they obtained in CCL2KO x 5xFAD mice and complement these by experiments performed on murine astrocytes and assessing concurrently the impact of LPS and amyloid beta.

The other way around, another option would be to use a rat model of acute neuroinflammation induced by the peripheral administration of LPS. 

Round 2

Reviewer 2 Report

The authors did not adress the issues I raised.

I agree with the authors that a short report format would be ok but, still, the content is not only quantitatively poor but overall the quality and relevance of these data are not really convincing.

There is yet no convincing explanation on the rationale behind switching from in vivo experiments of the cortices of CCL2KO x 5xFAD mice to in vitro experiments measuring the effects of LPS + CCL2 on the synthesis of IL-1 beta in rat cultured astrocytes. Neither the choice of LPS instead of amyloid beta nore the choice of rat astrocytes instead of murine astrocytes are reasonably justifiable.

With regard to GPR32, a pseudogene in rodents, I do not agree with the following author's statement: "The altered expression of pseudogenes, while not producing proteins, can be used as a reference to detect the regulation of the corresponding human genes". Pseudogenes exert effects on their own mostly via their transcribed RNA (when transcribed)  and there is poor evidence that their regulation is mediated by mechanisms similar to those regulating the functional couterparts of such pseudogenes. Moreover, assessing the regulation of a pseudogene by RT-PCR is not a reliable method (PMID: 33110161)

To circumvent this problem, why not having used human astrocytes instead of rat astrocytes? GPR32 is not a pseudogene in humans and there are now many commercial sources of human astrocytes 
